# High Variation in Protist Diversity and Community Composition in Surface Sediment of Hot Springs in Himalayan Geothermal Belt, China

**DOI:** 10.3390/microorganisms11030674

**Published:** 2023-03-07

**Authors:** Peng Zhang, Jie Xiong, Nanqian Qiao, Shuai Luo, Qing Yang, Xiaodong Li, Ruizhi An, Chuanqi Jiang, Wei Miao, Sang Ba

**Affiliations:** 1Laboratory of Wetland and Catchments Ecology in Tibetan Plateau, Faculty of Ecology and Environment, Tibet University, Lhasa 850000, China; 2Key Laboratory of Aquatic Biodiversity and Conservation, Institute of Hydrobiology, Chinese Academy of Sciences, Wuhan 430072, China

**Keywords:** Himalayan geothermal belt, protist, diversity, community composition, 18S rRNA gene amplicon

## Abstract

Hot springs are some of the most special environments on Earth. Many prokaryotic and eukaryotic microbes have been found to live in this environment. The Himalayan geothermal belt (HGB) has numerous hot springs spread across the area. Comprehensive research using molecular techniques to investigate eukaryotic microorganisms is still lacking; investigating the composition and diversity of eukaryotic microorganisms such as protists in the hot spring ecosystems will not only provide critical information on the adaptations of protists to extreme conditions, but could also give valuable contributions to the global knowledge of biogeographic diversity. In this study, we used high-throughput sequencing to illuminate the diversity and composition pattern of protist communities in 41 geothermal springs across the HGB on the Tibetan Plateau. A total of 1238 amplicon sequence variants (ASVs) of protists were identified in the hot springs of the HGB. In general, Cercozoa was the phylum with the highest richness, and Bacillariophyta was the phylum with the highest relative abundance in protists. Based on the occurrence of protist ASVs, most of them are rare. A high variation in protist diversity was found in the hot springs of the HGB. The high variation in protist diversity may be due to the different in environmental conditions of these hot springs. Temperature, salinity, and pH are the most important environmental factors that affect the protist communities in the surface sediments of the hot springs in the HGB. In summary, this study provides the first comprehensive study of the composition and diversity of protists in the hot springs of the HGB and facilitates our understanding of the adaptation of protists in these extreme habitats.

## 1. Introduction

Hot springs are some of the most special environments on Earth. Microorganisms (prokaryotic and eukaryotic) in hot springs play a critical role in shaping geothermal ecosystems [1,2,3,4,5]. The high temperatures, high levels of dissolved minerals, and high levels of acidity in these environments have resulted in the evolution of a wide range of adaptations by the resident microorganisms [6,7,8]. These adaptations make the hot springs a valuable resource for exploring microbial diversity and understanding its distribution patterns. In recent years, high-throughput sequencing has been increasingly utilized to study microbial communities in various environments and ecosystems, such as marine, river, soil, and geothermal spring ecosystems [8,9,10]. As geothermal springs are unique sites for extremophilic microorganisms, studies of the microorganisms in these systems have been essential to our understanding of the diversity and evolution of life on Earth [2,8,11]. In the past decade, the diversity of microorganisms in hot spring ecosystems has been studied, mainly with a focus on the prokaryotes and fungi [12,13]; such studies have been conducted on hot springs in Yellowstone National Park [14], the Taupō Volcanic Zone in New Zealand [8], Tengchong in China [3,15], the Himalayan region in Pakistan [2], the Great Basin in the United States [16], Kamchatka in Russia [17], Malaysia [18] and the Himalayan region in China [4,5,19,20]. These studies have provided important insights into the diversity and adaptation mechanisms of microorganisms in these extreme environments. However, these studies have intrinsic limitations. Firstly, they primarily concentrate on prokaryotic microorganisms or fungi communities, with little to no research on protists. Secondly, although some studies have been conducted on protists, the limited number of samples collected from hot springs prevents a comprehensive investigation of the diversity of these protists. Protists are a large and diverse group of eukaryotic microorganisms, including protozoa, unicellular algae, and fungi [21,22]. Protists play an important role in various ecological ecosystems, including freshwater, marine, soil, and extreme environments [9,23,24]. Using high-throughput sequencing, the diversity, composition and function of protists have been reported in many marine, river, and soil environments [25,26]. However, the protist diversity in hot springs is much less studied, especially in the surface sediment of hot springs [8]. Therefore, the protist diversity and community composition of hot springs are far from clear. The HGB is one of the most hot-spring-rich regions in the world [13]. The HGB systems are profuse and diverse, constitute one of the important geothermal zones in the world, and include some of the world’s largest thermal pools (e.g., Yangbajing geothermal field, South Western China). Geothermal springs as a representative extreme habitat, are not only characterized by high temperatures, but also exhibit various geochemical gradients [3,27], such as variable ionic strength, a large pH range, and enriched mineral elements, indicating that multiple stressors exist within geothermal environments [3]. These distinguishing features have led to a faster evolution of individual populations in extreme habitats than in normal environments, accelerating the turnover speed at the community level [28]. Until now, very little is known about the protist diversity in geothermal springs from this highly elevated region.

Here, we used 18S rRNA gene amplicon sequencing to (1) carry out the comprehensive investigation of the diversity, composition, and spatial distribution patterns of protist communities in the hot springs of the HGB; (2) determine the effects of environmental factors on the niches and diversity of protist communities of the hot springs in the HGB; (3) evaluate the relative impact of environmental factors (e.g., temperature and pH) on shaping the protist communities of the hot springs in the HGB; and (4) investigate the characteristics of protist communities in extreme conditions such as high-temperature environments. Elucidating these questions about protists in hot springs is crucial to gain a complete understanding of the adaptability of protist communities in geothermal ecosystems. The results of this study provide a comprehensive understanding of the ways in which protists are able to adapt to their environments in geothermal systems, and could also be a vital addition to global biogeographic diversity information.

## 2. Materials and Methods

### 2.1. Study Area and Sample Collection

Surface sediment samples from geothermal hot springs were collected throughout the HGB of Southern Tibet (26°20′~30°00′ N, 82°10′~95°20′ E), including the Lhasa, Shannan, Rikaze, Nagqu, Qamdo, and Nyingchi regions (Appendix A). Locations of the sampling sites are shown in Figure 1. In total, 123 surface sediment samples were collected from 41 hot springs in 2020. Samples were collected from 4 sites in Lhasa city (sites 1–4), 21 sites in Rikaze city (sites 5–25), 10 sites in Shannan city (sites 26–35), and 6 sites in Nyingchi city (sites 37–41). Each site was sampled 3 times, resulting in 123 samples in total. A 50 mL sample of surface sediment was collected from the top layer (approximately 0–20 cm) at each site and immediately frozen in cryotubes with liquid nitrogen. The samples were stored at −80 °C until DNA extraction.

### 2.2. Measurement of Environmental Parameters

Environmental factors of hot spring surface sediment samples were measured at each site; longitude (Lon), latitude (Lat), and altitude (ALT) were measured with a portable GPS locator (NAVA, F70, Beijing, China); electrical conductivity (EC) and salinity (SAL) were measured using a portable conductivity tester (HANA, HI993310, Padova, Italy); temperature (T) and pH were measured using a portable soil pH/temperature tester (HANA, HI99121, Padova, Italy); and the water content (Wg) of the surface sediment was measured with a soil-water tester (SIAS, SYS-SF, Dandong, China).

### 2.3. DNA Extraction, PCR, and Sequencing

DNA extraction from hot spring surface sediment samples was performed using a MoBio Power Soil DNA isolation kit (MoBio Laboratories, Carlsbad, CA, USA) according to the manufacturer’s protocol. Agarose gel electrophoresis (1% concentration) was used to check the quality of extracted DNA. The DNA concentration and purity were determined using a NanoDrop 2000 spectrophotometer (Thermo Fisher Scientific, Wilmington, DE, USA). The 18S rDNA V9 hypervariable region was amplified by polymerase chain reaction (PCR) using primer set 1391F (5′-GTACACACCGCCCGTC-3′) and 1510R (5′-TGATCCTTCTGCAGGTTCACCTAC-3′) with unique barcode sequences at both 5′ ends [29]. The PCR amplification was carried out in a 50 μL reaction system containing 5 μL 10× PCR buffer, 1.5 μL dNTP mixture (10 mM for each), 1.5 μL forward and reverse primers (10 μM), 0.5 μL Taq DNA Enzyme (TaKaRa), 2 μL of template DNA (10–30 ng), 1 μL of BSA, and 37 μL ddH_2_O. The PCR program was as follows: 98 °C for 60 s, 30 cycles of 98 °C for 10 s, 50 °C for 30 s and 72 °C for 30 s, and final extension at 72 °C for 5 min. The PCR products were sequenced using the PE250 strategy on the Illumina NovaSeq 6000 platform by Novogene (Beijing, China).

### 2.4. Bioinformatic and Statistical Analysis

Amplicon sequence variants (ASVs) were obtained using the DADA2 plug-in in QIIME2 software [30,31] and the ASV abundance table, and they were annotated with the SILVA database (version 138) [32]. Non-protist ASVs and low-abundance ASVs (<10 reads) were removed. To eliminate the influence of the sequencing depth on downstream analyses, 12,210 reads were randomly resampled for each sample. To minimize errors, the three replicate samples were averaged. Rarefaction curves were then generated for the normalized data. Representative sequences were aligned by MAFFIT plug-in in QIIME2, and a phylogenetic tree was constructed by FastTree plug-in in QIIME2. Alpha diversity indices (including richness index, Shannon–Wiener diversity index, Pielou evenness index, Simpson dominance index and Faith’s phylogenetic diversity (PD)) were calculated using the “vegan” package in R. Principal coordinate analysis (PCoA) and permutational multivariate analysis of variance (PERMANOVA) were performed based on the Bray–Curtis distance using the “vegan” package in R [33]. Mantel tests were used to determine correlations between environmental variables and selected characteristics of protist composition in the “linkET” package in R. The “betapart” package [34] in R was used to calculate and partition the total beta-diversity for all sampling sites to understand the relative contribution of species spatial turnover or replacement (Repl) and species loss or nestedness (Nest) to overall beta-diversity, as defined by Baselga [35]. Sampling sites were mapped in ArcMap 10.8.1.

## 3. Results

### 3.1. High Variation in Environments and Diversity of Protists in the HGB

Six environmental factors affecting the surface sediment samples in the hot springs from the HGB were measured, and the results are shown in Figure 2A. Overall, the temperature ranged from 16.60 °C (site 22) to 76.10 °C (site 10), with a 4.6-fold difference. The pH ranged from 6.30 (site 22) to 10.18 (site 10), with a 1.6-fold difference. The electrical conductivity ranged from 0.27 (site 22) to 4.88 (site 10), with an 18-fold difference. The salinity ranged from 0.07 ppt (site 22) to 0.92 ppt (site 10), with a 13-fold difference. The water content ranged from 28.53 (site 38) to 52.57 (site 8), with a 2-fold difference (Appendix A). These results show that the environments were highly varied among the hot springs in the HGB. Among these six environmental factors, unsurprisingly, the salinity and electrical conductivity were highly correlated. However, no significant correlations were found between any two other environmental factors (Figure 2A). These results highly suggest that there are high variations among the environments of hot springs in the HGB. When we modeled the spatial distributions of the environmental factors using a kriging interpolation method, the pH and altitude were higher for the northwest than for the southeast hot springs, and the salinity and electrical conductivity were higher for the southeast than for the northwest hot springs; however, this trend was not observed for the temperature and water content (Figure 2B).

A high variation of environments in hot springs should precipitate a high variation in protist diversity. We investigated the protist diversity using 18S rRNA gene amplicon. About 8.5 million high-quality reads were obtained from 41 hot spring samples. Overall, the sequencing of 10,000–50,000 V9 rDNA reads was sufficient to approach saturation in protist richness for most samples. Rarefaction curves plateaued for the majority of samples, indicating good sampling of the protist community (Appendix A). A total of 3292 ASVs were identified as belonging to eukaryotes using all of the sequencing reads. Among them, 1238 ASVs belong to the protists (e.g., eukaryotic algae and protozoa) and 566 ASVs could be annotated at the family level. The Shannon, richness, and phylogenetic diversity indexes were used to estimate the alpha diversity of the protists in different hot springs (Figure 2B). The richness index varied from 8 to 307, with a mean of 90. The phylogenetic diversity index (PD) varied from 2.64 to 48.61, with a mean of 20.16. The Shannon diversity index varied from 0.23 to 4.96. These results suggest high variations of protist alpha diversity in these hot springs. Meanwhile, we estimated protist beta-diversity based on Bray-Curtis, Jaccard, and UniFrac distance (Figure 2D). For the beta-diversity, Bray Curtis distance varied from 0.29 to 1.00, with a median of 0.97. The Jaccard distance varied from 0.50 to 1.00, with a median of 0.94. The UniFrac distance varied from 0.31 to 0.96, with a median of 0.69. The beta diversity also indicates the high variation of protist diversity. In summary, the diversity of protists in the hot springs of the HGB is highly varied, corresponding to the high variation in their environments.

### 3.2. The Global Picture of Protist Community in Surface Sediment of Hot Springs

We first investigated the protist community by combining all the samples collected from the hot springs of the HGB. The results showed that the proportion of the ASV’s number was higher than 1% in 17 taxa, including Cercozoa (17.53%), followed by Bacillariophyta (14.46%), Ochrophyta (9.85%), Ciliophora (9.53%), Discosea (6.54%), Chlorophyta (6.54%), Euglenozoa (6.30%), Bigyra (5.90%), Tubulinea (5.25%), Apicomplexa (3.39%), Cryptophyta (2.10%), and Evosea (1.45%) (Figure 3A).

For the abundance, 12 taxa showed relatively high abundance (>1%): Bacillariophyta (26.92%), Ciliophora (16.03%), Ochrophyta (12.12%), Chlorophyta (11.26%), Cercozoa (9.84%), Tubulinea (5.64%), Discosea (5.37%), Apicomplexa (3.00%), Euglenozoa (2.57%), Bigyra (1.71%), Evosea (1.62%), and Cryptophyta (1.07%) (Figure 3B). In summary, Ochrophyta and Tubulinea showed both a high number of ASVs and their abundance. Bacillariophyta and Ciliophora showed high abundance but had a low number of ASVs. Cercozoa and Bigyra showed a high number of ASVs but had a low abundance. Then, we checked the protist community in different hot springs. We found that the protist community differentiates extensively in different hot springs, even at the phylum level (Figure 3C).

Based on the occurrence of ASVs in hot springs, we classified all the ASVs into four categories: (1) constant ASVs—occurrence higher than 70%; (2) common—occurrence ranged from 30% to 70%; (3) occasional—occurrence ranged from 10% to 30%; and (4) rare—occurrence lower than 10%. Only 0.16% of the ASVs were found to be constant, which mainly included the ASVs of Bacillariophyta and Tubulinea. The majority of the ASVs belonged to the occasional and rare category, 14.46% and 82.71%, respectively (Figure 3D). Regarding the occurrence of ASVs in different taxa (Figure 3E), Chlorophyta, Cryptophyta, and Evosea were only found in the occasional and rare categories. Cercozoa, Discosea, Euglenozoa Ochrophyta, Bigyra, Apicomplexa, and Ciliophora were only found in the rare and constant categories. Tubulinea and Bacillariophyta were only found in the constant category.

According to the beta diversity, the protist community in the 41 sample sites could be generally classified into five clusters: Clusters 1 to 5 (C1, C2, C3, C4, and C5) (Figure 4A). The differences in protist communities among the five clusters were investigated using principal coordinate analysis (PCoA) based on Bray Curtis distance. The results showed that the PCoA1 and PCoA2 explained 13.5% and 8.4% of the total variation in protist community, respectively (Figure 4B). PERMANOVA analysis showed that the composition of protist communities in the surface sediment of the hot springs had significant spatial variation (*p* < 0.05). In C1, the predominant phylum of the protists was Ciliophora (21.23%), followed by Chlorophyta (18.46%) and Ochrophyta (16.22%). In C2, the predominant phylum was Bacillariophyta (44.28%), followed by Cercozoa (10.89%) and Ochrophyta (9.02%). In C3, the predominant phylum of the protists was Bacillariophyta (33.56%), followed by Cercozoa (12.68%) and Ciliophora (11.82%). In C4, the predominant phylum was Bacillariophyta (26.18%), followed by Chlorophyta (17.25%) and Ciliophora (16.52%). Lastly, in C5, the predominant phylum was Ciliophora (25.16%), Bacillariophyta (17.31%), and Chlorophyta (14.64%) (Figure 4C).

Beta diversity can reflect two different phenomena, the spatial turnover or replacement (Repl), and the species loss or nestedness (Nest) of assemblages, which result from two antithetic processes. The Repl is caused by changes in species abundance, and the Nest is caused by the loss of species. Both Repl and Nest are important concepts for understanding the spatial distribution and variation of beta diversity. We used the Jaccard method to calculate and partition total beta diversity to understand the relative contribution of Repl and Nest to overall beta diversity. The results of the relative contribution for Repl and Nest show that in protist communities, the mean contributions of the turnover pattern results from the Repl and Nest components to the beta diversity were 47.80% and 42.04%, respectively (Figure 4D). The results show that the turnover and nestedness component jointly affected the species composition difference (beta diversity) among communities.

### 3.3. Protists with High Relative Abundance in Hot Springs of the HGB

We checked the protist ASVs in the hot springs of the HGB with relatively high abundance. Table 1 lists the top 20 ASVs with the highest relative abundance by combining all samples. The results show that most of them belong to the groups Alveolata and Stramenopiles, including the genus *Peridinium*, *Polytoma*, *Melosira*, and *Paramecium*. One thing we noted is that the protist community in the hot springs is relatively simple, with one or several dominant species having a high relative abundance—up to 50% of the total protist reads (Figure 5A). This trend is especially evident in high temperature (>45 °C) or alkaline (pH < 7) hot springs (Figure 5B).

We then checked the protists with a relative high abundance in the high-temperature hot springs (i.e., >45 °C). A total of 12 hot springs showed a temperature higher than 45 °C. Figure 5C showed the accumulative relative abundance of the top five ASVs in each of these 12 high-temperature hot springs. In general, the dominant protists in these 12 hot springs are highly different. Protists from the phylum of Bacillariophyta, Ciliophora, Apicomplexa, Amoebozoa (Discosea, Evosea, and Tubulinea), Ochrophyta, and Cercozoa can be found in these 12 hot springs. At the genus level, *Breviata*, *Ascogregarina*, *Eimeria*, *Monocystis*, *Encyonema*, *Melosira*, *Navicula*, *Nitzschia*, *Peridinium*, *Sellaphora*, *Thalassiosira*, *Sorodiplophrys*, *Paracercomonas*, *Rhogostoma*, *Polytoma*, *Anteholosticha*, *Paramecium*, *Spirostrombidium*, *Urosoma*, *Cryptomonas*, *Endostelium*, *Korotnevella*, *Petalomonas*, *Filamoeba*, *Schizoplasmodium*, *Chrysochromulina*, *Chlamydomyxa*, *Ochromonas*, *Paraphysomonas*, *Spumella*, and *Vermamoeba* showed a relatively high abundance in these hot springs, suggesting that these kinds of species evolved the ability to adapt to the high temperature.

### 3.4. Relationship between Environmental Factors and Protist Communities in Hot Springs of the HGB

Pearson’s correlation coefficients were used to evaluate the relationships between the protist diversity index and environmental factors (Table 2). The ASV richness had significant negative correlations with electrical conductivity (*R* = −0.48, *p* < 0.01), salinity (*R* = −0.46, *p* < 0.01), temperature (*R* = −0.36, *p* < 0.05), and altitude (*R* = −0.55, *p* < 0.01). The Shannon index had significant negative correlations with electrical conductivity (*R* = −0.36, *p* < 0.05), salinity (*R* = −0.34, *p* < 0.05), and altitude (*R* = −0.44, *p* < 0.01). The ACE index had significant negative correlations with electrical conductivity (*R* = −0.54, *p* < 0.01), salinity (*R* = −0.52, *p* < 0.05), temperature (*R* = −0.41, *p* < 0.01), and altitude (*R* = −0.54, *p* < 0.01).

Linear regression analysis was performed in order to clarify the relationship between environmental factors distance and beta diversity of protists in the surface sediment of hot springs (Figure 6A). The results show that the distance of the temperature and water content are significantly and positively correlated with beta diversity, indicating that these two environmental factors affected the diversity of the protist communities in the hot springs. Meanwhile, mantel tests were used to explore the effect of environmental factors on variation of protist communities in different clusters of hot spring. The protist communities showed significant correlations with temperature in C1-C4, electrical conductivity and salinity in C4, and altitude in C5 (Figure 6B). These results suggest that temperature is the most important environmental factor, which affects the protist communities in the surface sediments of the hot springs in the HGB.

Understanding the niche breadth of individual species is crucial to better predict the potential outcomes of factor change on the species distribution in an environment. The niche breadth is usually calculated as the standard deviation of each variable (environment factor), weighted by the relative abundance of ASVs at each sampling site. In order to explore the habitat niche breadth of protists for each ASV along the gradient of environmental factors, we analyzed the niche breadth of the protist ASVs across all sampling sites of six major environmental gradients. Slight similarities in the niche breadth pattern of protists were observed for electrical conductivity, salinity, and pH gradient. The niche breadth of protists also positively correlated with electrical conductivity, salinity, and pH (Figure 6C). However, the niche breadth showed a unimodal relationship with temperature and altitude. the niche breadth increased with temperatures up to 40 °C and then decreased as the temperature increased further. Similarly, the niche breadth increased with altitudes up to 3800 m and then decreased as the altitude increased further. In addition to this, the water content did not have a significant effect on the niche breadth of protists. In general, altitude and temperature seem to be the limiting factors for the niche breadth of protists in the surface sediment of hot springs in the HGB.

## 4. Discussion

### 4.1. Protists Are Widely Distributed in the Sediment of Hot Springs in HGB

This study is the first molecular investigation of protists in the surface sediment of hot springs in the HGB. The protist diversity and composition pattern in the surface sediment of hot springs in the HGB were studied using a 18S rRNA gene amplicon. The 41 hot springs are highly variable in their environmental factors, such as temperature, pH, and salinity (Figure 2A). This study performed the first comprehensive measurement of the physical and chemical parameters in hot springs in the HGB, providing basic data on the environment for future research. Although there have been some previous studies on the hot springs in the region, they only looked at a limited number of hot springs [4,5,19,20]. Additionally, we found that some hot springs have special conditions, such as high levels of calcium and sulfur ions. Although the exact ion concentrations were not measured, the presence of large quantities of calcium and sulfur compounds was observed at the sample sites. These hot springs will provide a unique environment to study the adaptation of microorganisms to extreme environments, including searching for microorganisms capable of utilizing calcium and sulfur in the future.

We found that the protist alpha diversity and beta diversity are highly variable in these hot springs (Figure 2C,D). These findings are consistent with the findings in the geothermal hot springs of New Zealand, which showed that a high variation of environments in hot springs drive a high variation of protist diversity [8]. In addition, the limitations of amplicon sequencing can lead to an overestimation of the protist diversity, and the ability to determine the absolute abundance is not possible; only the relative abundance can be determined [36,37,38]. In our study, 1238 ASVs of protists could be identified by the SILVA database. These results suggest that geothermal environments harbor an unexpected protist diversity [8,39]. However, many of the ASVs of protists are rare. The most widespread ASV was only found in 70% of the hot springs. This means a high variation in the protist diversity among these hot springs, and this may be due to a high environmental difference. These results are very different from previous findings in the rivers and lakes of the Tibetan Plateau [22], which showed that most protist ASVs are widely distributed. For the composition of protist communities, we found that Bacillariophyta and Ciliophora seemed to be the most dominant (i.e., the relative abundance is the highest) phyla in hot springs in the Tibetan Plateau (Figure 2C). This finding is consistent with previous studies on protist communities in river and lake environments in the Tibetan Plateau [22]. The Amoebozoan clade containing the Discosea thermogram was of interest, as it appears to be a diverse taxon with an evolutionary adaptation to withstand high temperatures [8,40]. However, we found that Cercozoa was the phylum with the highest richness of surface sediment in hot spring samples in the HGB, which made up 16.65% on average. This finding is consistent with previous studies on protist communities in sediment samples [9,41]. Of these protists taxa, most are free-living, such as Cercozoa, Bacillariophyta, Ochrophyta, Ciliophora and Amoeba [42]. Cercozoa and Bacillariophyta are frequently found in sediment samples [43]. Amoebas are characterized by their ability to change shape and move by extending pseudopods [44]. We also found some parasitic protists in the hot springs, such as Apicomplexa, which live with their hosts (metazoan or protozoan) [45]. In summary, we found that diverse protist taxa are widely distributed in the sediment of hot springs in HGB.

### 4.2. Adaptation to High-Temperature Environments of Many Protists

An interesting finding is that many protists have been identified in high-temperature hot springs (i.e., >45 °C). At the genus level, *Paramecium* was found in these high-temperature hot springs (Figure 5C). It is not possible that this is a result of contamination, as Jiang et al. isolated a *Paramecium* species through molecular and morphological investigation in high-temperature (46 °C) hot springs in the HGB [46]. However, whether *Paramecium* exists in high-temperature environments is still not clear; further experimental evidence is needed to verify this result. *Acanthamoeba* was found in these hot springs and has been identified using phenotypic and genotypic methods in hot springs in Chile [37]. In addition, *Trimyema* was also found in these hot springs and has been reported in high-temperature environments in Vulcano Island [47,48]. These free-living amoebas are widely distributed in extreme environments such as temperature, pH, and salinity. This may be due to their ability to form cysts, allowing some of these protists, such as *Trimyema*, to survive in these environments as extremophilic or extremotolerant organisms. Some studies reported that cysts of some ciliates remain viable for many years in a desiccated state and that they can survive dry heat up to 120 °C for a short period [42,47,48]. Thus, our results suggest that cyst formation may be an important strategy to adapt to high-temperature hot springs. *Ochromonas* was found in hot springs in the HGB, which may be adapted to environments with low pH levels (Figure 5B), and this is consistent with the findings of Schmidtke et al. [49]. In addition, the genera *Naegleria*, *Filamoeba*, *Acanthamoeba* and *Naegleria* have been found in surface sediment samples of hot springs in the HGB. The Echinamoebida is an extremely thermophilic amoeba that had been found in hot springs in Agnano Terme (Italy), Yellowstone National Park (USA), Kamchatka (Russia), and the Arenal Volcano (Costa Rica) [50]. We also found the presence of some genera of Bacillariophyta in these hot springs (Figure 5C). This may be due to the fact that Bacillariophyta have protective shells, which are favorable for DNA preservation. Thus, their DNA may have been detected in the sediment samples [51]. Furthermore, *Eimeria* and *Ascogregarina* as parasites were found in the hot springs, possibly together with their metazoan or protozoan hosts [45,52].

We found that the protist community of hot springs is relatively simple in high-temperature (>45 °C) hot springs. Additionally, the species composition of the protists in the 41 hot spring sampling sites lacks distinct patterns. This result is different in lakes and rivers, which usually see more similar protist composition at the phylum level. However, this result is similar to the finding of prokaryotes in the Trans-Himalayan Plateau, which also showed high variation in the composition of prokaryotes [53,54]. These phenomena might be explained by intense environmental pressures in these extreme hot springs, which limit the growth of protists [7,55]. Previous studies have shown that abiotic factors (e.g., temperature and pH) affect the microbial diversity and community composition in hot springs [56,57,58]. In addition, many studies also revealed that the temperature could lead to a difference in the metabolism and growth of microorganisms [8,49]. To identify the environmental factors affecting the protist niche breadth, we performed habitat niche analysis. We found that temperature and altitude are important factors in determining the niche breadth of protists. This may be explained by the significant correlation between altitude/temperature and other environmental factors such as pH and salinity (Figure 6B), which were previously reported to affect the protist niche breadth [25,59]. In addition, temperature could also enhance the dissolution of mineral elements and accelerate the rate of redox reactions [60,61], which may influence the resource availability to microorganisms. Therefore, temperature could directly and indirectly (i.e., interacting with minerals) shape the sediment community composition.

## 5. Conclusions

In summary, this study provides the first comprehensive investigation of protist diversity and composition using molecular techniques in the geothermal ecosystems of the HGB. Our findings suggest that hot springs have high levels of protist diversity, even in hot springs with extreme conditions (i.e., temperatures greater than 45 °C), indicating the adaptation to the extreme environments of some protists. Temperature, salinity, and pH are the most important environmental factors affecting the protist communities in the surface sediments of the hot springs in the HGB. This study may be a valuable contribution to advance the understanding of the global biogeographic diversity of protists.

## Figures and Tables

**Figure 1 microorganisms-11-00674-f001:**
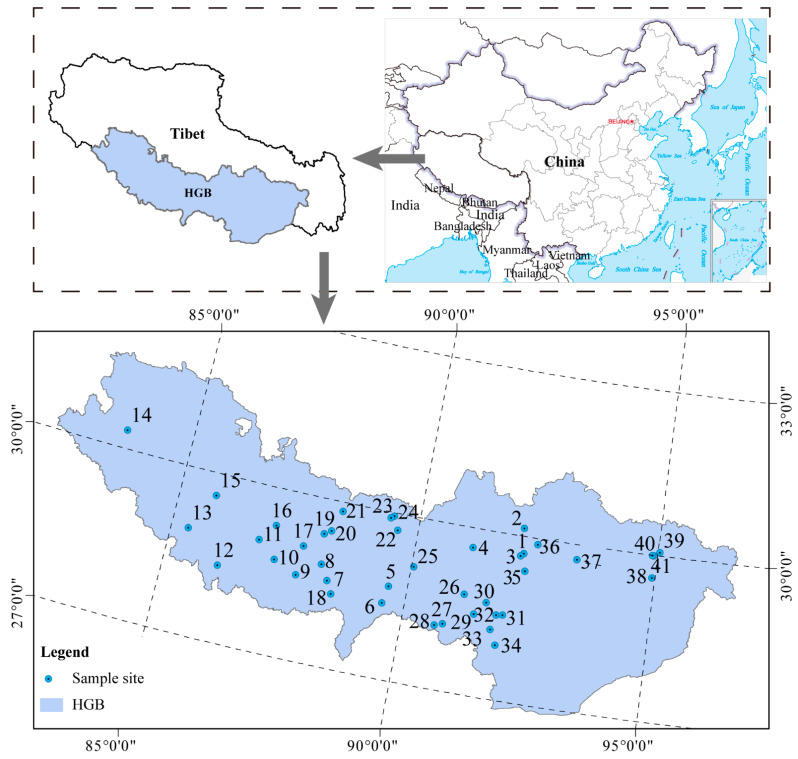
Distribution of the 41 surface sediment sampling sites in the HGB.

**Figure 2 microorganisms-11-00674-f002:**
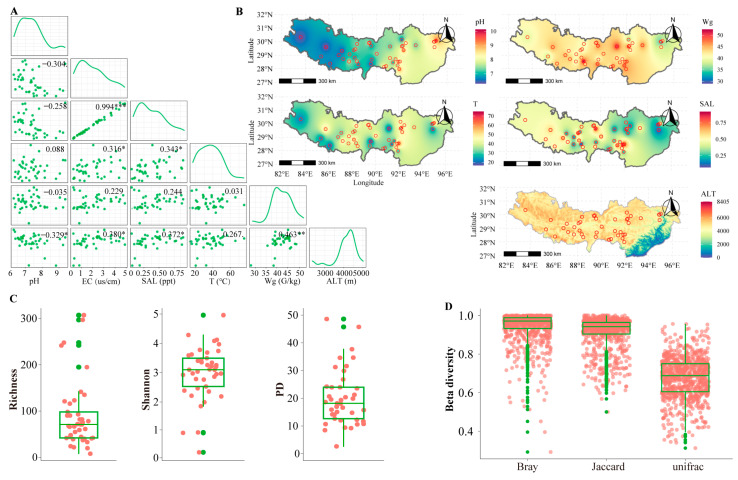
High variation in environmental factors and diversity of protists. (**A**) Spearman correlations among environmental factor parameters. (**B**) The spatial patterns of environmental factors. (**C**) Richness, Shannon, and PD index of protists. (**D**) Beta diversity of protists. Notes: *** *p* ≤ 0.001; ** *p* ≤ 0.01; * *p* ≤ 0.05; other relationships are not statistically significant.

**Figure 3 microorganisms-11-00674-f003:**
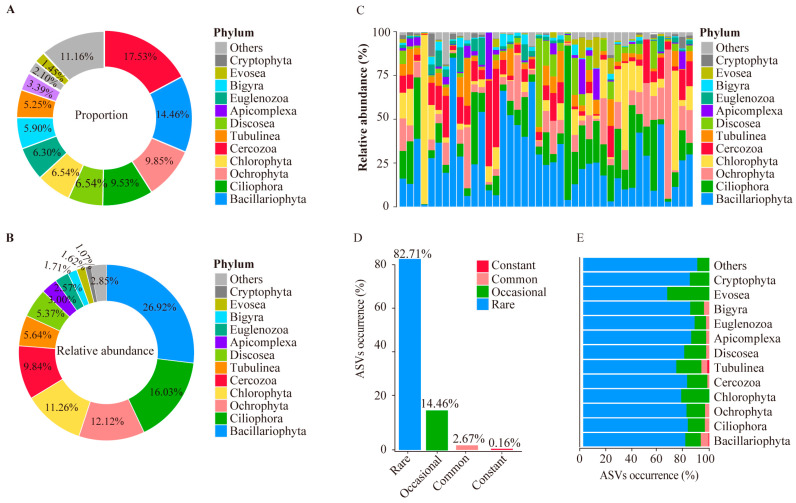
Features of protist community in hot springs in the HGB. (**A**) Percentage of ASVs in different taxa of protists. (**B**) Relative abundance of ASVs in different taxa of protists. (**C**) Relative abundance of ASVs in different taxa per sample. (**D**) Distribution of ASV abundance of protists. (**E**) Distribution of ASV count of protists. (**D**) Occurrence of ASVs in protist communities. (**D**) Occurrence of ASVs in protist communities in different taxa.

**Figure 4 microorganisms-11-00674-f004:**
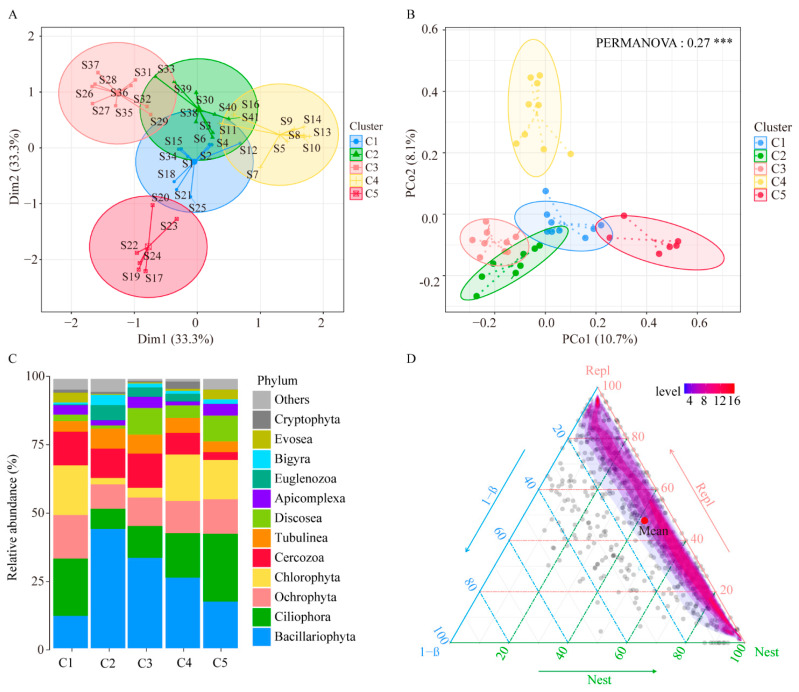
Beta diversity of protists. (**A**) Cluster analysis of protist communities in all sample sites. (**B**) Principal coordinate analysis (PCoA) and analysis of PERMANOVA of protist communities in the five clusters. (**C**) Relative abundance of the main phyla of protists in the five clusters. (**D**) The triangular plot of protist communities made by using the Sørensen index. Notes: *** *p* ≤ 0.001.

**Figure 5 microorganisms-11-00674-f005:**
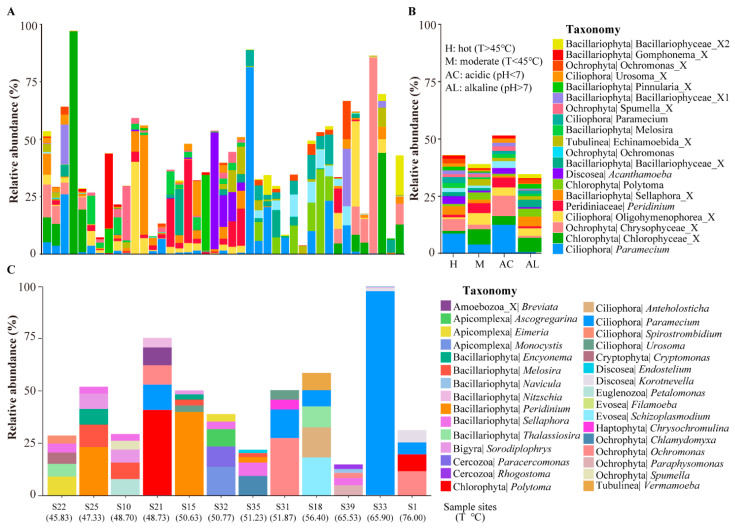
Protists with relatively high abundance in hot springs of HGB. (**A**) The relative abundance of the top 20 ASVs in all sample sites. (**B**) The relative abundance of the top 20 ASVs in hot (H: T > 45 °C), moderate (M: T < 45 °C), acidic (AC: pH < 7) and alkaline (AL: pH > 7) springs. (**C**) The accumulative relative abundance of the top five genus in each of 12 high temperature hot springs.

**Figure 6 microorganisms-11-00674-f006:**
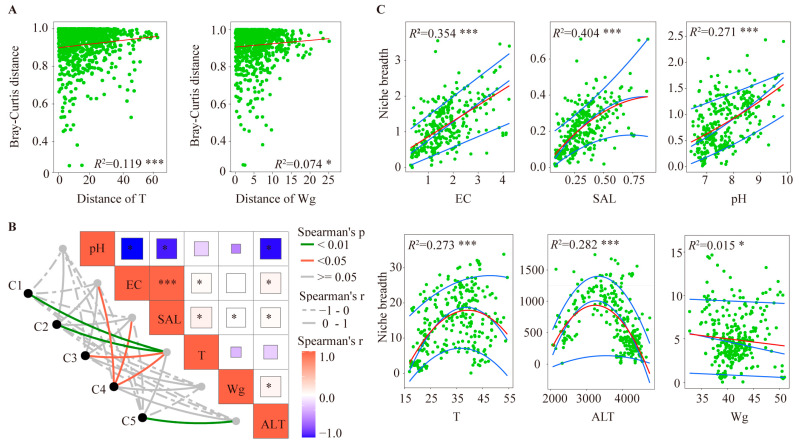
Relationship between environmental factors and protist community. (**A**) Relationship between Bray Curtis distance of protist communities and Euclidean distance of environmental factors. (**B**) Environmental drivers of protist communities, as evaluated by Mantel tests in the different clusters of hot springs. (**C**) Habitat niche breadth analysis of protists for each ASV along the gradient of environmental factors. Red lines indicate linear regression. Blue lines from bottom to top indicate the quantile regressions of 10%, 50%, and 90%, respectively. Notes: *** *p* ≤ 0.001; * *p* ≤ 0.05; other relationships are not statistically significant.

**Table 1 microorganisms-11-00674-t001:** Top 20 ASVs with relatively high abundance by combining all samples.

ASV Id	Relative Abundance	The Best Taxonomic Match
ASV 731	0.052	Alveolata|Ciliophora|Oligohymenophorea|Peniculida|Parameciidae|*Paramecium*|*Paramecium tetraurelia*
ASV 596	0.051	Archaeplastida|Chlorophyta|Chlorophyceae_X
ASV 1055	0.038	Stramenopiles|Ochrophyta|Chrysophyceae_X
ASV 675	0.034	Alveolata|Ciliophora|Intramacronucleata|Oligohymenophorea_X
ASV 218	0.029	Stramenopiles|Bacillariophyta|Dinophyceae|Peridiniales|Peridiniaceae|*Peridinium*|*Peridinium balticum*
ASV 90	0.026	Stramenopiles|Bacillariophyta|Bacillariophyceae|Naviculales|Sellaphoraceae|*Sellaphora*_X
ASV 570	0.021	Archaeplastida|Chlorophyta|Chlorophyceae|Chlamydomonadales|Chlamydomonadaceae|*Polytoma*|*Polytoma uvella*
ASV 867	0.020	Amoebozoa|Discosea|Longamoebia|Centramoebida|Acanthamoebidae|*Acanthamoeba*|*Acanthamoeba lenticulata*
ASV 208	0.015	Stramenopiles|Bacillariophyta|Bacillariophyceae_X1
ASV 1010	0.013	Stramenopiles|Ochrophyta|Chrysophyceae|Chromulinales|Chromulinaceae|*Ochromonas*|*Ochromonas*_X1
ASV 1159	0.013	Amoebozoa|Tubulinea|Echinamoebida_X
ASV 211	0.013	Stramenopiles|Bacillariophyta|Coscinodiscophyceae|Melosirales|Melosiraceae|*Melosira*|*Melosira varians*
ASV 730	0.013	Alveolata|Ciliophora|Oligohymenophorea|Peniculida|Parameciidae|*Paramecium*|*Paramecium tetraurelia*
ASV 1015	0.016	Stramenopiles|Ochrophyta|Chrysophyceae|Chromulinales|Chrysocapsaceae|*Spumella*_X
ASV 110	0.011	Stramenopiles|Bacillariophyta|Bacillariophyceae_X2
ASV 67	0.011	Stramenopiles|Bacillariophyta|Bacillariophyceae|Naviculales|Pinnulariaceae|*Pinnularia*_X
ASV 746	0.009	Alveolata|Ciliophora|Spirotrichea|Sporadotrichida|Oxytrichidae|*Urosoma*_X
ASV 1033	0.009	Stramenopiles|Ochrophyta|Chrysophyceae|Ochromonadales|Chromulinaceae|*Ochromonas*_X2
ASV 53	0.009	Stramenopiles|Bacillariophyta|Bacillariophyceae|Cymbellales|Gomphonemataceae|*Gomphonema*_X
ASV 177	0.009	Stramenopiles|Bacillariophyta|Bacillariophyceae_X3

**Table 2 microorganisms-11-00674-t002:** Pearson’s correlation coefficients (*R*) between the alpha diversity index and environmental factors.

Index	pH	EC	SAL	T	Wg	ALT
Richness	0.25	−0.48 **	−0.46 **	−0.36*	−0.14	−0.55 **
Shannon	0.29	−0.36 *	−0.34 *	−0.12	−0.16	−0.44 **
Simpson	0.18	−0.25	−0.23	−0.01	−0.05	−0.19
Pielou	0.08	−0.11	−0.08	0.16	0.03	−0.02
ACE	0.27	−0.54 **	−0.52 **	−0.41 **	−0.10	−0.54 **

Notes: ** *p* ≤ 0.01; * *p* ≤ 0.05; other relationships are not statistically significant.

## Data Availability

The data presented in the study are deposited in the National Genomics Data Center repository, accession number PRJCA015299.

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
