# Peer review of "High Variation in Protist Diversity and Community Composition in Surface Sediment of Hot Springs in Himalayan Geothermal Belt, China"

_microorganisms, 2023, doi:10.3390/microorganisms11030674_

Round 1
Reviewer 1 Report
The manuscript by Zhang et al. describes the diversity of protists in 41 geothermal springs across the Himalayan Geothermal belt in Tibet discovered by the 18S rRNA gene metabarcoding. There is a number of publications devoted to the prokaryotes of the geothermal ecosystems, but the manuscript by Zhang et al. is one of the first works where the protist component of the microbial communities in hot springs is in the main focus. I found the manuscript to be rather interesting, and in general my opinion about it is rather positive.
The experiments were fulfilled in three replicates, which is necessary to obtain a quality data. The methods applied to analyze the sequencing data are correct and comprehensive. The results looks scientifically sound and quite interesting.
There are numerous grammar mistakes, which become the most abundant in Discussion. Many of them are minor, but they tend to accumulate during reading. I tried to correct some of them while reading (see below), but I suggest that the authors ask for a language check.
Despite the overall positive impression, the manuscript needs to be revised, and below the authors may find a number of questions to address before the manuscript can be accepted.
I have a question concerning sampling: why did the authors take the sediment but not water form the sampling sites? Sediment may accumulate DNA coming from different sources, also from extinct organisms, which may influence the results.
I also wonder if the primers used could also detect some metazoans or higher plants DNA.
I think that Discussion would benefit if the authors give some attention to several points:
- metabarcoding is not a reliable quantitative approach, as all primers may have and do have certain bias. So when it comes to abundance estimates, it should be mentioned that it is not necessary absolutely precise parameter;
- line 241 - the authors conclude that more harsh conditions (high temperature and alkaline pH) lead to less number of the protist taxa in the hot spring and also to the fewer dominants. It should also be mentioned in Discussion as a simple but important conclusion.
- the manuscript lacks a bit more of general biological description of the protist taxa (at least phyla) discovered by metabarcoding in the geothermal springs. At least it is necessary to mention that some phyla preferentially include parasitic organsims, while the other phyla consist of free-living organisms; that Bacillariophyta found to be one of the dominants are diatoms, so they may be better adapted to survive or their DNA may be better protected by a frustule and remains longer in sediment even without the alive cells?
For the rest comments, please go along the list of my remarks below.
Line 32 – ‘in this extreme habitat‘- please put in plural
Line 36 – ‘on the earth planet’ – either on Earth, or on the planet
Lines 43-44 – I don’t agree that it was widely studied
Line 45 – “Protist is a large and diverse group” – please change for “protists are”
Line 46 – and also unicellular fungi (if the algae are mentioned separately); moreover, all algae are eukaryotic, so I suggest “including protozoa, unicellualr algae, and fungi”
Lines 63-65 “Elucidating the diversity and composition of protists in hot springs is essential to reach a comprehensive understanding of protist community adaptability in geothermal ecosystems on Tibetan Plateau.” – I suggest the authors not to limit the outcome of the study as important just for Tibetan Plateau. I think that it is more important that their results are essential for a better understanding of protist adaptation to survive in these extreme biotopes, and also for a comprehensive understanding of the whole microbial communities (including also prokaryotes) function in geothermal ecosystems.
Line 67 – please change here and throughout the text “18S amplicon” for “the 18S rRNA gene amplicon”
Line 69-70 – please delete “reveal the diversity and spatial distribution of protists in this ecosystem and”
Section 2 is traditionally named “Materials and methods”, not vice versa
Line 76 please put “Locations” in plural
Line 96 – I did not find in the internet the Gene DNA Extraction Kit (Novogene, China) description and protocol, and I wonder how does it work with 50 ml of the sediment sample. Could the authors please describe this part of procedure in more detail?
Lines 99-101 The primers need to have a reference.
A question: why V9 region of the 18S rRNA gene was chosen, but not, for example, V4 region, which is known as a suitable marker for metabarcoding?
Lines 129-144 I would prefer if the authors do not use the abbreviations for the environmental factors but, instead, write the complete words (temperature, salinity, water content, etc.). It would not make the article much longer, but it would be easier to follow. It becomes especially annoying in section 3.4.
Line 139 “any two environmental factors” – please add ‘other’ after ‘two’
Line 145 “should expect” please change for “allowed to expect”
Lines 158-161 – could the authors please explain the difference between Bray-Curtis, Jaccard, and UniFrac distance, why does it make sense to use all three methods?
Section 3.2:
In the first paragraph there is clearly a mistake with the Figures number (should be Fig. 3, instead it is Fig. 4 throughout). Also there is a discrepancy in percentages between the text and the legend of Figure 3A.
Lines 185-186 – what is the relation of the five clusters on Fig. 3C to the lake and the river?
Figure 3 legend – please change “ASVs number in .. different taxa of protists” for “the ASVs corresponding to the different taxa of protists” (ASVs cannot be ‘in’ the taxa, each ASV is an amplicon obtained from a representative of a certain taxon).
Lines 220-228 – it would be useful if the authors explain the meaning of the terms “spatial turnover or replacement (Repl) and species loss or nestedness (Nest) of assemblages”. These parameters will be not clear to many readers.
Table 1 – why some ASVs are redundant? For example, ASVs 730 and 731 are both assigned to Paramecium tetraurelia.
Line 250 – Eimeria and Ascogregarina are a bit surprising in the list of the thermotolerant organisms. Both genera belong to Apicomplexa and are known to include only parasites, but not free-living organisms. How can it be that the ASVs corresponding to them were found in the extreme environment? Could it be that they appeared in the system together with their metazoan (or protozoan) host?
Lines 256-263 is a part of Discussion
Line 321 please change “is driving” for “drives”
Line 325 “However, many of ASVs are hot spring specific protists.” is a terrible sentence, as ASVs are not protists, they correspond to certain protists.
Line 328 please change ‘difference’ for ‘different’
Lines 252, 342 – please delete the extra space
Lines 347-348 “It is not possible a contamination, because both the molecular and 347 morphology investigation are found this species.” – 1) please rephrase, as the sentence is incorrect; 2) it is the first time that the authors mention some morphological investigation. Does it mean that they also screened the samples for living protists and identified some of those? If so, it has to be mentioned in ‘Methods’ section.
Line 350 – While Trimyema indeed make cysts, Paramecium is not able to encyst. So this hypothesis in regards of Paramecium fails, even if in general I agree that encystment can be a strategy helping to survive in the adverse conditions.
In general, presence of Paramecium in the hot springs leaves a lot of questions. First, for Paramecium temperature over +35 C degrees is a fatal heat shock, and at +35 degrees only some thermotolerant mutants are able to survive. So finding of ASV assigned to this genus in the hot spring with temperature 66 degrees is highly unlikely.
Then Paramecium is not a ciliate which may thrive in the sediment, it usually stays closer to the surface and needs a lot of bacteria as food to survive. Again, in case of anaerobic ciliate Trimyema its finding in the sediment of a hot spring is not that surprising, as it fits to the known life style of this ciliate.
Also Paramecium sequence finding in the samples from the hot springs in New Zealand is not discussed in the paper by Oliverio et al. 2018, it is just present on one of the figures in that paper together with some sequences of other obviously non-thermophilic ciliates.
If possible, I would like to check myself the ASVs that the authors identified as belonging to Paramecium tetraurelia, or at least I would like to ask the authors to double-check this assignement. Could it be that those ASVs belong to some other Peniculid ciliate?
Line 352 Please put commas where necessary;
Reference 22 please correct the format.
Reviewer 2 Report
The article is interesting and could make an important contribution to the field, but unfortunately in its current form the manuscript lacks research depth, visible by a focus on the case study rather than the research issue, proved by poor introduction, discussions, and lack of conclusions. Thus, the manuscript requires a strong development of these sections. In general, the manuscript presents too many results, without a clear indication of their contribution to the advancement of the field. Moreover, the article lacks the international exposure required for publication in an international journal. Detailed comments are provided for each section of manuscript.
The research goals (lines 67-71) are rather modest and point more to the case study and not to a scientific issue, lacking research depth. This is in line with the very sketchy introduction, based on only 18 references. In its current form, the introduction fails to define a research framework and even lesser to identify the shortcomings (ambiguities, controversies, misconceptions or lacks) of previous studies, justifying the need for research, and emphasizing the novel and original elements of the current study. The introduction should be developed to identify a research issue; the authors should include studies on the micro-biota of other thermal springs beyond China, because China is not the only country in the world.
Figure 1 shows the inability of authors to write up research. This is an article for an international journal, and not a report for the national authorities. The authors should present a map showing the location of the study area in an international context, making visible the neighboring countries with their names, so that a Brazilian researcher could understand it too. China is not the only country in the world!
The most important section of a research article, the Discussions, is missing. What the authors call "Discussions" is only a repetition of the main results. The section is meant to emphasize the importance of research, justifying its publication. Normally, this section includes include (A) the significance of results - what do they say, in scientific terms; (B) the inner validation of results, against the study goals or hypotheses; (C) the external validation of results, against those of similar studies from other countries, identified in the literature; (D) the importance of results, meaning their contribution (conceptual or methodological) to the theoretical advancement of the field; (E) a summary of the study limitations and directions for overcoming them in the future research. The "Discussions" should be rewritten entirely.
Conclusions are not sufficiently broad in scope, and lack research depth, pertaining only to the case study and being in fact just a summary of the main findings. Conclusions are meant to deliver a scientific message, far away beyond the case study, to the entire scientific community, making a clear contribution to the theoretical (conceptual or methodological) development of the field. Conclusions must be developed beyond the case study.
The abstract looks like a shopping list, focusing on the case study only, and not on the broader implications of research and only on what has been done, without the slightest indication on why it has been done, and what knowledge gap is actually being filled in. The abstract is supposed to deliver ideas, and not state the research steps in brief and provide useless figures instead of their significance. It needs to be rewritten entirely, and shift the focus from the case study to the research issue investigated in the study.
Round 2
Reviewer 1 Report
I appreciate the work the authors did to revise the manuscript. They carefully addressed the criticisms and the comments, and now the article is significantly improved. The language issue is also fixed in a proper way. I am grateful for the correct explanations of some terms and statistical methods applied for the data analysis. I am also convinced by the authors' replies on my questions. There are still some minor things to correct and fix, but when this is done, the manuscript can be accepted for publication.
Please check my remarks with the line numbers according to the pdf file of the tracked version, as this was the only file available to me:
Line 39 - what does the key word "Composition" mean?
Line 80 - you misunderstood my previous remark, I meant that either you say "on Earth" or you say "on the planet".
lines 104-108 - when I asked to add a general biological description of the protist taxa, I supposed this would be done in Discussion. If you do it in Introduction, it looks a bit naive (and the questions follow: why did you select these taxa but not some others? what is the connection between amoebae locomotion and the geothermal springs? and so on). So I suggest again to transfer some important highlights about the protist taxa that were found in the research to the Discussion, there it would look much better.
Line 123 - "until now" and "currently" are synonyms, please delete one of them
line 126 - please change "into" for "of"
line 129 - is not it the same, environmental factors and environmental conditions?
line 132 - please change "to gaining" for "to gain"
line 238 - please change "18S amplicon" for "18S rRNA gene amplicon"
(line 486 - also please do the same instead of "18S rDNA amplicon")
line 264 "theof" is the typo
Figure 3D - please check the legend on figure itself, "coonon" and "occasiinal" are weird typos
line 317 - I suggest to include the explanation of Repl and Nest phenomena in the body of the manuscript, as the authors formulated it well in their rebuttal (The spatial turnover or replacement (Repl) and species loss or nestedness (Nest) of assemblages describe different aspects of beta-diversity, which is a measure of the change in species composition between different locations. "Spatial turnover or replacement (Repl)" refers to the difference in species composition between locations that are not caused by species loss. "Species loss or nestedness (Nest)" refers to the difference in species composition between locations caused by the loss of species in some locations but not in others.") It will not take too much space, and the terms are not so common.
Table 1 - I understood the authors explanation about why some different ASVs were assigned to the same organism. To make it more clear for the readers, I suggest to change in the headline of the table "Taxonomy" for "The best taxonomic match"
line 497 - please delete "ions"
line 509 - please change "difference" for "different"
lines 528-532 - please rephrase a bit, as now it seems as if you attribute Trimyema (a ciliate) to free-living amoebae
line 535 - is it really true that someone proved survival of Trimyema cysts at 120o degrees? hard to believe.
The last suggestion: a week ago there was an alert about the new publication in Ecologies (MDPI) by Malygina et al. that is closely related to the topic of this manuscript: https://doi.org/10.3390/ecologies4010009. There are just a few papers that are devoted to the occurrence of protists in the geothermal springs, so I would like to drag the attention of the authors to that article and I suggest to use it for the Discussion. I think the results desrcibed there strengthen the findings described in this manuscript.
Reviewer 2 Report
The authors addressed my previous requests, improving their manuscript.
Author Response
We thank the comments of the reviewer.